# Potential Therapeutic Effect of Micrornas in Extracellular Vesicles from Mesenchymal Stem Cells against SARS-CoV-2

**DOI:** 10.3390/cells10092393

**Published:** 2021-09-12

**Authors:** Jae Hyun Park, Yuri Choi, Chul-Woo Lim, Ji-Min Park, Shin-Hye Yu, Yujin Kim, Hae Jung Han, Chun-Hyung Kim, Young-Sook Song, Chul Kim, Seung Rok Yu, Eun Young Oh, Sang-Myeong Lee, Jisook Moon

**Affiliations:** 1Department of Biotechnology, College of Life Science, CHA University, Seongnam 13488, Korea; librorum0601@gmail.com (J.H.P.); love7855@chauniv.ac.kr (Y.C.); lim9030@chauniv.ac.kr (C.-W.L.); park20090724@gmail.com (J.-M.P.); yssong@chamc.co.kr (Y.-S.S.); bio.ckim501@gmail.com (C.K.); 2Paean Biotechnology, Incorporation, Daejeon 34028, Korea; sryu2@paeanbio.com (S.-H.Y.); yjkim@paeanbio.com (Y.K.); chkim@paeanbio.com (C.-H.K.); 3Research and Development Center, Green Cross WellBeing Corporation, Seongnam 13595, Korea; hjhan@gccorp.com; 4Division of Biotechnology, College of Environmental and Bioresources, Jeonbuk National University, Iksan 54596, Korea; fbtmdfhr159@naver.com (S.R.Y.); rwae00@naver.com (E.Y.O.); 5College of Veterinary Medicine, Chungbuk National University, Cheongju 28644, Korea; smlee@chungbuk.ac.kr

**Keywords:** SARS-CoV-2, COVID-19, miRNA, mesenchymal stem cell, extracellular vesicle, cytokine storm

## Abstract

Extracellular vesicles (EVs) are cell-released, nanometer-scaled, membrane-bound materials and contain diverse contents including proteins, small peptides, and nucleic acids. Once released, EVs can alter the microenvironment and regulate a myriad of cellular physiology components, including cell–cell communication, proliferation, differentiation, and immune responses against viral infection. Among the cargoes in the vesicles, small non-coding micro-RNAs (miRNAs) have received attention in that they can regulate the expression of a variety of human genes as well as external viral genes via binding to the complementary mRNAs. In this study, we tested the potential of EVs as therapeutic agents for severe acute respiratory syndrome coronavirus 2 (SARS-CoV-2) infection. First, we found that the mesenchymal stem-cell-derived EVs (MSC-EVs) enabled the rescue of the cytopathic effect of SARS-CoV-2 virus and the suppression of proinflammatory responses in the infected cells by inhibiting the viral replication. We found that these anti-viral responses were mediated by 17 miRNAs matching the rarely mutated, conserved 3′-untranslated regions (UTR) of the viral genome. The top five miRNAs highly expressed in the MSC-EVs, miR-92a-3p, miR-26a-5p, miR-23a-3p, miR-103a-3p, and miR-181a-5p, were tested. They were bound to the complemented sequence which led to the recovery of the cytopathic effects. These findings suggest that the MSC-EVs are a potential candidate for multiple variants of anti-SARS-CoV-2.

## 1. Introduction

Severe acute respiratory syndrome coronavirus 2 (SARS-CoV-2) is the positive sense single-stranded RNA virus that causes coronavirus 2019 (COVID-19). SARS-CoV-2 is a variant of the coronavirus SARS-CoV, which is associated with severe acute respiratory syndrome. SARS-CoV-2 is the seventh coronavirus known to infect humans; the others are 229E, NL63, OC43, HKU1, MERS-CoV, and the original SARS-CoV [1]. SARS-CoV-2 is a Sarbecovirus (β-CoV strain B) with an RNA genome of about 30,000 bases. Since SARS-CoV-2 was identified as the causative agent of COVID-19, scientists have sought to understand the genetic composition of the virus and to find ways to effectively treat the infection. Many biotech companies have focused on the development of vaccines against COVID-19. Because multiple trials of safety and efficacy tests remain, the higher efficacy of vaccines should be awaited to prevent or reduce the viral infection. Under these circumstances, the development of therapeutic agents for COVID-19 is in progress, suggesting that two research tracks should be pursued: vaccines and drug development to overcome infectious diseases.

The RNA genome has a high rate of mutation, making RNA viruses very heterogeneous [1]. As in HIV and influenza, SARS-CoV-2 viruses contain RNAs as their genome and are frequently susceptible to mutation. New SARS-CoV-2 variants have been identified in a stream and have cause doubt in the development of vaccines and therapies. For example, antibody-based therapeutics developed against one variant must be redesigned after the virus mutates. Consequently, influenza vaccines need to be updated annually. Additionally, viral variants develop resistance to antiviral drugs, and mild virus variants become virulent spontaneously. Consequently, therapeutic and preventive agents that can cope with viral mutations are needed urgently. The heterogeneity of SARS-CoV-2 asks us to find new approaches, such as recognizing common regions of multiple variants and deactivating viral functions.

MicroRNAs (miRNAs) are small non-coding RNAs of 18–25 nucleotides (nts); these small molecules are important regulators of gene expression because they suppress messenger (m)RNAs. miRNAs regulate approximately 30–70% of human gene expression by matching their “seed” sequence with complementary mRNA [2]. miRNAs bind directly to viral RNA genomes or affect viral replication by mediating changes in the host mRNAs. Several miRNAs bind directly to a wide range of RNA viruses and modulate pathogenesis [3]. A perfect match between miRNA and the whole mRNA target sequence leads to RNA cleavage, although this is rare in mammals [4]. More often, exact seed sequence matching results in translational inhibition, followed by RNA degradation [5]. In addition, the seed sequence, which is a short seven-nucleotide region at the 5’ end of the miRNA, binds to a complementary sequence within the 3′ UTR of the invading viral RNA and results in the degradation of the target mRNA strand [4]. Trobaugh D.W. et al. reported that Eastern Equine Encephalitis Virus (EEEV) was inhibited by the interaction of miR-142-3p with the complementary miR-142-3p binding site in the EEEV 3′ UTR region. However, removal of its 3′ UTR led to a reduction in viral replication, indicating that the 3′ UTR in the viral genome has its own function in viral infection [3,6,7]. Secondly, it is known that the 3′ UTR region of the virus genome is relatively conserved and forms a conserved special secondary structure which is known to play an important role in virus replication as well as stabilizing RNA [8,9], further confirming the importance of targeting 3′ UTR of the virus. Given that there is a plausible mechanism that the complementary miRNAs can potentially target the virus and, above all, exert antiviral effects by mainly acting on the miRNA binding site within the conserved 3′ UTR, the targeting of 3′ UTR of the viral genome can be a key strategy for antiviral effects. Extracellular vesicles (EVs) are small membrane-bound bodies that contain cargoes such as nucleic acids, proteins, lipids, and metabolites [10]. Release of EVs is an important form of intercellular communication that plays roles in the physiological and pathological processes underlying multiple diseases. One of the most fascinating features of EVs is their ability to transfer miRNAs to recipient cells and regulate cell conditions [11]. For example, miRNAs delivered by EVs to both immune cells and other cell types repress or degrade target mRNAs in recipient cells [12].

The regulatory or regenerative functions of miRNAs contained in stem cell-derived EVs have attracted attention as a novel therapeutic approach [13]. EVs secreted from mesenchymal stem cells (MSCs) mediate tissue regeneration and help to repair damage in a variety of diseases, including cardiac ischemia, liver fibrosis, and cerebrovascular disease. Additionally, MSC-EV has direct and indirect antimicrobial and immunomodulatory effects [14]. Multiple studies report the therapeutic effects of miRNAs delivered to host cells by MSC-EVs. MSC-EVs transport proteins, non-coding RNAs (ncRNA), miRNAs, and lipids, which mediate cardiac tissue repair and regulate the environment around damaged tissue, induce angiogenesis, promote proliferation, and prevent apoptosis [15]. When SARS-CoV-2 attaches to the ACE2 receptor to invade a host cell, immune cells such as T lymphocytes and dendritic cells (DCs) secrete and absorb EVs containing miRNAs that attack the infected viral RNA [16]. Thus, delivery of miRNA by MSC-EVs represents a new therapeutic mechanism for combating SARS-CoV-2 and regulating the pro-inflammatory environment.

Here, we identified candidate therapeutic miRNAs with important roles in the biological functions of virus-infected host cells, and characterized the antiviral effects of miRNAs derived from placenta-derived MSC-EVs (pMSC-EVs) or placenta EVs, which exhibit potent regenerative and anti-inflammatory effects. Five selected miRNAs (miR-92a-3p, miR-103a-3p, miR-181a-5p, miR-26a-5p, and miR-23a-3p) blocked SARS-CoV-2 RNA replication and suppressed virus-mediated pro-inflammatory responses by human bronchial epithelial cells and lung fibroblasts, all of which express ACE2 receptors. These effects were also observed in human brain cells and brain immune cells. Interestingly, we found that ACE2 is expressed in human fetal brain stem cells, suggesting that the brain could be a target of SARS-CoV-2.

Most importantly, we found that the five miRNAs bound to the 3′ UTR of SARS-CoV-2, the sequence of which is conserved among coronaviruses.

## 2. Materials and Methods

### 2.1. Cell Culture

Neuronal stem cells (NSCs) were generated from the central nervous system tissue of spontaneously aborted fetus from ectopic pregnancy during gestational week (GW) 8. Isolation method is described in Moon et al. [17]. Normal human placentas (≥37 GW) showing no evidence of medical, obstetrical, or surgical complications were obtained. Sample collection and use for research purposes were approved by the Institutional Review Board (IRB, 2009-06-074 and 2015-08-130) of Bundang CHA General Hospital (Seongnam, Korea).

Human lung fibroblast cell line LL24, human bronchial epithelial cell line Beas-2B, mouse microglial cell line BV2, and human neuroblastoma cell line SK-N-BE(2)C were obtained from ATCC (Manassas, VA, USA). LL24 cells were grown in RPMI-1640 medium (Gibco, Carlsbad, CA, USA) with 10% fetal bovine serum (FBS, Gibco) and 1× penicillin/streptomycin (P/S, Gibco). Beas-2B cells were cultured in 1× defined keratinocyte serum-free media (SFM, Gibco) with 5 µg/mL gentamycin (Gibco). BV2 cells were cultured in Dulbecco’s modified Eagle’s medium (D-MEM, Gibco) with 10% FBS and 1% P/S. Neuronal stem cells (NSCs) were cultured at 37 °C/5% CO_2_, and 3% O_2_ in DMEM/F12 (Gibco) with B27 supplement (Gibco), 50 µg/mL gentamicin, human bFGF, human EGF 20 ng/mL (PeproTech, Rocky Hill, NJ, USA), tocopherol, and tocopherol acetate (1 µg/mL; Sigma-Aldrich, St. Louis, MO, USA). SK-N-BE(2)C cells were cultured in DMEM with 10% FBS, 1X P/S. The cells were pre-treated with EVs for 24 h and then stimulated with LPS (LL24: 2 μg/mL, Beas-2B: 4 μg/mL, BV2 and NSCs: 1 μg/mL) for either 6 or 1 h.

### 2.2. EV Isolation and Nanoparticle Tracking Analysis (NTA)

EVs were isolated by size-exclusion chromatography (SEC) columns (Izon science qEV, Izon Science Ltd., Burnside, Christchurch, New Zealand). Samples were overlaid on the qEV size exclusion column, followed by elution with particle-free phosphate-buffered saline (PBS). The flow was collected in 500 µL fractions, and EV-rich fractions 7 and 10 were combined for analysis. The combined fractions were centrifuged at 1000× *g* for 1 min in an Ultrafree 0.22 μm centrifugal filter device (Merk Millipore, Burlington, MA, USA).

For NTA analysis, EVs were diluted in PBS and examined under a ZetaView Nanoparticle Tracking Video Microscope (Particle Metrix, Inning, Germany). For each measurement, three cycles were performed by scanning 11 cell positions and capturing 60 frames per position. The following settings were used: focus, autofocus; camera sensitivity for all samples, 80.0; shutter, 100; scattering intensity, 1.2; cell temperature, 23 °C. After capture, the videos were analyzed using the built-in ZetaView Software 8.02.31 (Particle Metrix, Inning, Germany) with the following parameters: maximum particle size, 1000; minimum particle size, 5.

### 2.3. EVs Staining

For fluorescence imaging, EVs were biotinylated using EZ-Link Sulfo-NHS-LC-Biotin (Thermo Fisher Scientific, Waltham, MA, USA). Biotinylated EVs were loaded onto Zeba spin desalting columns (7K MWCO; Thermo Fisher Scientific) to remove the remaining free biotin. Additionally, then, 20 μL of biotinylated EVs were placed in a streptavidin-coated glass slide (Arrayit Corporation, Sunnyvale, CA, USA). After 30 min, EVs were fixed with BD Fix Perm (BD Biosciences, San Jose, CA, USA) and blocked with 0.2% BSA. Immunofluorescence staining was performed using anti-human CD63 (1:100; SC-5275, Santa Cruz Biotechnology, Dallas, TX, USA) for 2 h at room temperature (RT), followed by incubation with Alexa Fluor 488-conjugated secondary antibodies (1:500; Invitrogen, Carlsbad, CA, USA) for 1 h at RT. Additionally, then, these slides were mounted and analyzed by confocal fluorescence microscopy (Leica, Wetzlar, Germany).

For PKH26 staining, EVs were labeled with the PKH26 Red Fluorescent Cell Linker Kit for General Cell Membranes (Sigma-Aldrich).

For TEM imaging, 5 µL aliquots of diluted sample containing 0.5 µg protein were dropped onto hydrophilic grids. After a few minutes, the grids were washed with distilled water and contrasted for 20 s with 2% uranyl acetate. Images were acquired on a JEM-1010 microscope (JEOL, Seoul, Korea) operating at 80 kV.

### 2.4. Western Blotting

EVs and cells were lysed in 10× RIPA buffer containing a protease inhibitor cocktail tablet (Roche, Basel, Switzerland) and phosphatase inhibitors II and III (Sigma-Aldrich). An aliquot of each sample was subjected to 10% SDS-PAGE, followed by Western blotting. Blots were blocked for 1 h in 10% skim milk. Antibodies specific to the following proteins were obtained from the indicated suppliers: CD81 (1:1000, Santa Cruz Biotechnology), CD9 (1:1000, Santa Cruz Biotechnology), CD63 (1:1000, Santa Cruz Biotechnology), TSG101 (1:500, Abcam, Cambridge, UK), Vinculin (1:2000, Abcam), p65 (1:1000, Cell Signaling Technology, Danvers, MA, USA), phospho-p65 (1:1000, Cell Signaling Technology), and GAPDH (1:10,000, Santa Cruz Biotechnology). Primary antibodies were incubated with blots overnight at 4 °C, followed by incubation with anti-rabbit or anti-mouse horseradish peroxidase (HRP)-conjugated secondary antibodies (Jackson ImmunoResearch, West Grove, PA, USA) for 1 h. Immunoreactivity was detected using enhanced chemiluminescent HRP substrate (Merck Millipore, Burlington, MA, USA).

### 2.5. Immunocytochemistry (ICC)

Cells were fixed with 4% paraformaldehyde (PFA) for 30 min. Cells were washed with PBS and blocked with 1% BSA for an additional 20 min at RT. Cells were then incubated with anti-NF-κB p65 subunit (1:50, Santa Cruz Biotechnology) for 24 h at 4 °C. Following washing with PBS to remove excess primary antibody, the cells were further incubated with Alexa Fluor 488-conjugated goat anti-mouse IgG (1:2000; Invitrogen) for 1 h at RT, and stained with Gold Antifade reagent containing 40,6-diamidino-2-phenylindole (DAPI, Invitrogen) for 5 min. Subsequently, the slides were coverslipped and visualized using a fluorescence microscope (Nikon, Minato City, Tokyo, Japan).

### 2.6. RNA Extraction and Quantitative PCR (qPCR)

Total RNA for qPCR was extracted from cells and EVs using the TRIzol reagent (Invitrogen). Next, cDNA was synthesized from 1 μg of total RNA using a cDNA kit (Bioneer, Daejeon, Korea). For qPCR analysis, reaction mixtures contained 0.5 μM primer mixture, SYBR-Green with low ROX (BioFact, Daejeon, Korea), nuclease-free water (Ambion, Austin, TX, USA), and cDNA. Reactions were performed on a StepOne Real-Time PCR instrument (Applied Biosystems, Foster City, CA, USA). Quantification of gene expression was based on the CT value for each sample.

### 2.7. Antiviral Activity Assay

The African green monkey kidney epithelial cell line (Vero) was used as an appropriate cell line for SARS-CoV-2 propagation. The cells were cultured in DMEM (Gibco) with 10% FBS (Gibco). The antiviral activity assay was carried out using the NMC-nCoV02 strain of SARS-CoV-2. The virus was propagated in Vero cells and the titer of propagated viral stock was expressed as the tissue culture infective dose 50 (TCID_50_). All experiments using the virus were performed in a Biosafety Level 3 (BSL-3) laboratory located in College of Medicine and Medical Research Institute, Chungbuk National University (Cheongju, Korea) and the College of Environmental and Bioresources, Jeonbuk National University (Iksan, Korea). Cytotoxicity values of the EVs on Vero cells were determined using the methyl thiazolyl tetrazolium (MTT) assay.

For TCID_50_ assay, the SARS-CoV-2 samples were serially diluted 10-fold and 0.1 mL of each dilution was added to triplicate wells of 96-well plates containing confluent Vero cells (final volume 0.2 mL/well). After 3 days, cells were fixed by 10% formaldehyde solution for 5 min and stained by 1% crystal violet. The wells were inspected for the presence of virus, as judged by the appearance of cytopathic effect (CPE). The virus endpoint titration (dilution required to infect 50% of the wells) was expressed as TCID_50_/mL. CPE (%) was calculated, along with the virus titer (number of positive wells/total number of wells × 100).

The cytotoxicity inhibition assay was used to assess the antiviral activity of each EV. Briefly, Vero cells were grown in a 96-well plate until 80% confluent. Subsequently, the culture medium was removed from each well and 10^1^–10^2^ TCID_50_ of SARS-CoV-2 and different concentrations (2-fold serial dilutions) of EVs were added to each well. For the virus control, 10^1^–10^3^ TCID_50_ of virus plus the highest concentration of DMSO were added to six wells. Additionally, six wells were treated with DMSO alone (negative control). The plates were incubated and CPE monitored for up to 3 days post infection.

Additionally, the immunofluorescence assay was used to assess the antiviral activity of EVs. Vero cells were cultured and confluent cells were infected with SARS-CoV-2 at multiplicity of infection (m.o.i) of 0.01. After 1 h absorption, virus inoculum was removed and fresh medium containing EVs (1, 5, 25 μg/mL) was added into the chamber slide. Sixteen hours later, cell culture medium was removed, and cells were used for immunofluorescence assay.

### 2.8. Indirect Immunofluorescence Assay

Vero cells grown on 8-well chamber slides were fixed overnight at 4 °C with acetone and methanol mixture. Cells were washed with PBS, followed by blocking and permeabilization with 0.1% Triton-X 100 (Sigma-Aldrich) and 5% normal goat serum (NGS, Abcam) in PBS for 1 h at RT. After washing with 0.1% Triton-X 100, cells were incubated with anti-SARS-CoV-2 nucleocapsid antibody (SinoBiological, Beijing, China) and Alexa 488-conjugated goat anti-mouse IgG (Jackson Immunoresearch Laboratories, West Grove, PA, USA). Nuclei were stained with DAPI (Thermo Fisher Scientific). Stained cells were mounted with anti-fade mounting solution (Vectashield, Vector Laboratories, Burlingame, CA, USA) and observed by fluorescence microscopy (Olympus, Tokyo, Japan)

### 2.9. Transfection and Reporter Assay

A 208 bp fragment of the 3′ UTR of the SARS-CoV-2 genome was synthesized by Bionics. The 3′ UTR fragment was digested with Xba I and inserted downstream of the firefly luciferase gene in pGL3-control (Promega, Madison, WI, USA), yielding pGL3 SARS-CoV-2 3′ UTR-Luc. All constructs were verified by sequencing. The mutant 3′ UTR of the SARS-CoV-2 genome was generated by randomly designed sequence of the miRNA binding site and then inserted in pGL3 vector to make a pGL3 SARS-CoV-2 MUT-3′ UTR-Luc plasmid. The mutated sequence was as follows: 5′- TCTAGACAATCTTTAATCAGTGTGTAACATTAGGGAGGGTAGCTAAGAGCCACCACATTTTCACCGAGGCCACGCGGTTGCTACGTCCTAAGACAGTGAACAATGCTCACATTTTCACCGAGGCCACGCGGTTGCTACGTCCTAAGACAGTGAACAATGCTAGGGTATTGACGCTATATGGAAGAGCCCTAATGTGTAAAATTAATTTTAGTCCGAGCTACCTGCCTATTTTTTAATAGCTTCTTAGGAGAATGACTCTAGA-3′.

For transfection, SK-N-BE(2)C cells were cultured in 24-well plates (1.2 × 10^5^ cells/well) with antibiotic-free DMEM 1 day prior to transfection. Transfections were carried out with Lipofectamine 2000 (Invitrogen). To test particular miRNAs’ mediated suppression of luciferase activity, the cells were transfected with 0.2 µg of pGL3 SARS-CoV-2 3′ UTR-Luc and 0.3 µg of pRSV βgal as an internal control along with either 20 nM miRNA (hsa-miR-92a-3p, hsa-miR-26a-5p, hsa-miR-23a-3p, hsa-miR-103-3p, or hsa-miR-181-5p, Bioneer). Twenty-four hours after transfection, cells were lysed with 100 µL of lysis buffer (25 mM Tris-phosphate (pH 7.8), 2 mM DTT, 2 mM CDTA (1,2-diaminocyclohexane-*N*,*N*,*N*′,*N*′-tetraacetic acid), 10% glycerol, and 1% Triton X-100), followed by addition of an equal volume of firefly luciferase substrate. Then, the luciferase activity was measured in a luminometer plate reader and normalized against beta-galactosidase activity. In order to examine whether miRNAs downregulate luciferase activity in a sequence-specific manner, SK-N-BE(2)C cells were transfected with either WT or MUT 3′ UTR vector along with pMSC-EVs (0, 5, 10 μg), and then luciferase activity was analyzed.

### 2.10. RNA Sequencing and Data Processing

Small RNA sequencing was performed using extracellular vesicles obtained under 8 different conditions in pMSCs media and 2 placenta tissue extracts. Sequencing was performed at the Beijing Genomics Institution (BGI, Shenzhen, China) using BGISeq-500. Reads were aligned to a human reference genome (GRCh38) using subread aligner [18] and the featureCounts tool was used to obtain miRNA read counts [19]. The miRNA read counts were normalized, and miRNAs expressed at low levels were filtered using the edgeR R package. Expression of each miRNA was transformed to CPM. A read quality control was performed using qrqc in R package and the distribution of small RNAs was calculated using sRNAtoolbox [20]. Data regarding expression of ACE2 by human tissues and cell lines were obtained from The Human Protein Atlas database [21].

### 2.11. MiRNA Target Prediction and Functional Analysis of Binding Sites

The PITA tool [22] was used to investigate miRNA binding sites in the 3′ UTR. The SARS-CoV-2 complete genome sequence (NC_045512.2) was obtained from the NCBI reference sequence database [23] and the 3′ UTR sequence was extracted from the SARS-CoV-2 complete genome. The PITA tool was used to predict binding sites for miRNAs. The default values were used for PITA analysis. Unconventional miRNA binding sites were predicted by the miRDB custom prediction tool [24]. To determine the biological function of miRNAs, experimentally validated targets were obtained from the miRTarBase [25]. The miRNAs that made up a small percentage of the total reads (<1%) were excluded from the analysis. To investigate the biological function of miRNAs, GO term and KEGG pathway analysis were performed using the DAVID Bioinformatics Resource 6.8 [26]. The results of functional analyses were visualized using the Cytoscape 3.8 and Pathview R packages [27].

To investigate conserved regions within miRNA binding sites in SARS-CoV-2, the 3′ UTR sequences were obtained from the NCBI reference sequence database. The virus 3′ UTR sequences from the following coronaviruses were used: SARS coronavirus (NC_004718.3), SARS-CoV-2 (NC_045512.2), SARS coronavirus BJ01 (AY278488.2), Bat SARS coronavirus HKU3-1 (DQ022305.2), Bat SARS-like coronavirus SL-CoVZC45 (MG772933.1), Bat SARS-like coronavirus SL-CoVZXC21 (MG772934.1), and six complete genomes of SARS-CoV-2 from Korean patients (MT730002, MT678839, MT304474, MT304475, MT304476, and MT039890). The 3′ UTR sequences of the coronaviruses were aligned using the MUSCLE tool [28]. The default value was used for MUSCLE analysis.

### 2.12. Statistics

Statistical analysis was performed using either ANOVA, followed by Tukey’s HSD post hoc test, or a generalized linear model followed by a least square mean post hoc (R basic functions and lsmeans package). For the qPCR results, all statistics were based on 2^−∆Ct^ values. A graph of qPCR results for inflammation-associated genes shows fold change values. A heatmap of the log2-CPM of miRNAs in EVs was produced using the R package gplots. *p* values for GO term analysis and KEGG pathway analysis were corrected for multiple comparisons using the Benjamini–Hochberg method. Data are presented as the mean ± standard error of the mean. A *p* value or adjusted *p* value < 0.05 was considered significant.

## 3. Results

### 3.1. Profiles of MiRNAs of pMSC-EVs and Placenta EVs

To find the mechanism of EV’s antiviral effect, we first analyzed miRNAs within EVs. Based on the fact that the viral genome can be targeted by human miRNAs, we hypothesized that miRNAs within EVs directly interact with the SARS-CoV-2 genome. EVs obtained from eight MSCs under various cell culture methods and from six placenta derivatives were assessed by small RNA sequencing. Unassigned reads were most common, followed by miRNAs and tRNAs, indicating that miRNAs are the major cargo in EVs (Appendix A). The profiles of miRNA in EVs obtained from eight placenta MSCs and six placental derivatives EVs (PD-EVs) were very similar (Figure 1A). MSC-EVs miRNAs were profiled using small RNA sequencing and ranked (highest to lowest) in terms of counts per million (CPM) (Figure 1B). Read counts between biological replicates showed a strong correlation (Figure 1C). The top 18 miRNAs accounted for 80.1% of all EV miRNAs (Figure 1D). The remaining miRNAs were excluded from further analysis because they were present at very low read counts and comprised a very small percentage of the total reads (0.02–0.96%), and consequently were deemed unlikely to have a significant biological effect relative to the more abundant miRNAs. The number of genes targeted by the top 18 miRNAs and the number of targeted genes related to the inflammatory response are listed in Figure 1E. In addition, the number of inflammatory response genes targeted was further assessed and the results of enriched pathway analysis are also shown in Figure 1E. The majority of the genes contributed substantially to specific inflammatory responses related to cytokine–cytokine receptor interactions, TNF, NF-κB, chemokines, Toll-like receptors, and the Jak–STAT signaling pathway, indicating that their key function is immunomodulation (Figure 1E). The miRNAs affecting the most targets related to inflammatory responses were miR-92a-3p, miR-21-5p, miR-24-3p, miR-let-7b-5p, miR-let7a-5p, and miR-181a-5p. Moreover, the expression levels of 84 common miRNAs present in EVs derived from eight MSCs cultured under various conditions and six placental derivatives were very consistent across samples (Appendix A).

### 3.2. Antiviral Effect of Placenta-Derived EVs against SARS-CoV-2

EVs isolated from placenta stem cells (pMSC-EVs) were positive for the common EV marker CD63 (Appendix A). The hydrodynamic diameter of EVs measured by nanoparticle tracking analysis (NTA) was 121.8 nm (Appendix A) and a representative transmission electron microscopy (TEM) image exhibited the typical EV morphology (Appendix A). In addition, Western blotting detected typical EV markers, including CD81, CD63, TSG101, and CD9 (Appendix A). To verify the effect of EVs against SARS-CoV-2, we first examined the cytotoxicity of EVs in an MTT assay using Vero cells. Vero cells were treated for 24 h with two-fold serial dilutions of EVs (0.003–4.7 μg). No cytotoxicity was observed at any concentration tested and rather, cells proliferated after treatment with EVs at all doses (Figure 2A, *ps* < 0.05). Subsequently, we developed a 96-well plate assay in which live cells were stained with crystal violet, whereas cells showing virus-mediated CPE were not. To confirm the cytotoxic concentration of SARS-CoV-2, Vero cells were infected with the virus (10-fold serial dilutions: 10^1^ TCID_50_/well, 10^2^ TCID_50_/well, and 10^3^ TCID_50_/well, *n* = 3 replicates) and incubated for 3 days. The amount of cell death after 3 days was confirmed by examining CPE. Cell detachment was observed at doses of 10^1^ to 10^3^ TCID_50_/well (Figure 2B, left panel). Next, we examined the antiviral effects of EVs against SARS-CoV-2 at three different TCID_50_ concentrations. EVs at 5, 2.5, and 1.25 μg showed 100%, 100%, and 66% efficacy, respectively, at the titer of 10^1^ TCID_50_. EVs at 5 and 2.5 μg showed 66% and 33% efficacy, respectively, at a titer of 10^2^ TCID_50_. However, EVs did not show antiviral activity at a virus titer of 10^3^ TCID_50_ (Figure 2B, right panel). In addition, infection was significantly reduced in 5 and 25 μg (Figure 2C,D, *ps* < 0.001) as measured by immunofluorescence assay. To further test whether miRNAs in EVs directly mediate antiviral effects, SARS-CoV-2-infected cells were treated with 50, 200, and 350 ng of total miRNAs isolated from EVs, which is equivalent to 250 ng, 1 μg, and 1.75 μg of EVs, respectively. Infection was significantly reduced in 200 ng and 350 ng (Figure 2C,D, *ps* < 0.001). In order to understand the mechanism of antiviral effects mediated by either total EVs or miRNAs in EVs, miRNAs in EVs were analyzed and identified (Figure 3 and Table 1). Among the most abundant miRNAs in EVs, miR-92a-3p, miR-181a-5p, and miR-26a-5p are able to target the SARS-CoV2 with high efficiency and have the potential to regulate inflammatory responses. Furthermore, three miRNAs were individually treated and significantly reduced viral infection under all conditions (Figure 2C,D, *ps* < 0.001). Viruses infecting the respiratory system can invade cells expressing the ACE2 receptor, resulting in cell damage. Because ACE2 receptors are expressed primarily in the liver, kidney, male reproductive tissue, muscle, and the gastrointestinal tract (GI), these organs can be damaged by such viruses (Appendix A). Although expression of ACE2 in the brain has not been studied in detail and the Brain Atlas database suggests that expression is very low (Appendix A), clinical outcomes continue to suggest that the brain might be a target of SARS-CoV-2. We found that neuronal stem cells (NSCs) strongly express ACE2 receptors (Appendix A) at levels comparable with known virus target organs, such as LL24, indicating that the brain is a target organ for SARS-CoV-2. The ACE2 receptor gene was expressed in pMSCs at levels similar to those of lung cells (Appendix A). In addition, the ACE2 mRNA was also observed in pMSC-EVs, even at a much higher level compared to other cells including pMSCs (Appendix A). This result provides the possibility that pMSC-EVs can exert antiviral effects by including ACE2 in the membrane and acting competitively with SARS-CoV-2. To determine whether EVs regulate pro-inflammatory cytokine release in response to SARS-CoV-2, we examined their indirect antiviral effects on various cell types such as LL24, Beas-2B, and NSCs, including brain immune cells (microglia cells; BV2), expected to be targeted by SARS-CoV-2. EVs labeled with PKH26 were observed after 24 h. Based on the results, the next experiments were conducted using an EV treatment time of 24 h (Appendix A). Cells treated with EVs significantly reduced inflammation stimulated by lipopolysaccharides (LPS). Expression of IL-1β and IL-6 in LL24 and Beas-2B cells (Figure 2E,F, *ps* < 0.05), of IL-1β, IL-6, and TNF-α in BV2 cells (Figure 2G, *ps* < 0.01), and IL-1β and IL-6 in NSCs (Figure 2H, *ps* < 0.05) fell significantly, whereas expression of IL-1β, IL-6, and TNF-α in cells not pre-treated with EVs increased after exposure to LPS. These results suggest that EV treatment may prevent cytokine storms caused by SARS-CoV-2 infection. NF-κB is a major transcription factor stimulated by LPS. Under normal conditions, NF-κB is located within the cytoplasm. Upon activation, the IKK complex phosphorylates IκBα, leading to translocation of NF-κB to the nucleus. Although NF-κB is an important regulator of immune responses, we found that expression of mRNA encoding NF-κB was not affected by EVs after LPS stimulation. Therefore, we examined the translocation of NF-κB protein. NF-κB was translocated into nucleus in LPS-treated Beas-2B cells, but was sequestered in the cytoplasm after EV treatment (Figure 2I). In addition, expression of NF-κB phospho-p65 in Beas-2B cells increased upon treatment with LPS and decreased after treatment with LPS and EVs; however, total p65 expression was unchanged (Figure 2J,K). These results suggest that EVs regulate the inflammatory responses by modulating the NF-κB signaling pathway and p65 translocation.

### 3.3. Direct Viral Effect of MiRNAs in EVs on SARS-CoV-2

To explore the possibility that miRNAs within EVs interact with the SARS-CoV-2 3′ UTR, we investigated whether the SARS-CoV-2 3′ UTR contains potential miRNA binding sites. We aligned the 3′ UTR sequences of SARS-CoV-2 isolates (Figure 3A) and analyzed the alignment using PITA software. Table 1 shows information about binding sites in the SARS-CoV-2 genome and the results of binding prediction analyses. PITA software identified 17 miRNAs predicted to target the 3′ UTR of SARS-CoV-2 (Table 1) and 28 miRNAs with sequences expected to bind to whole genome sites within the SARS-CoV-2 (Appendix A). Five miRNAs, miR-92a-3p, miR-26a-5p, miR-23a-3p, miR-103a-3p, and miR-181a-5p, out of seventeen miRNAs were selected based on the thermodynamic energy score and high scores in two prediction tools: PITA and miRDB. Each of the candidate sites was assigned a logistic probability as a measure of confidence. We obtained the total free energy of each miRNA, based upon the fact that the lower the binding thermodynamic energy (kcal/mol), the stronger the binding. For example, the binding energy of miR-181a-5p for the 3′ UTR was −18.7 kcal/mol, suggesting that binding of miRNA to the 3′ UTR binding would proceed spontaneously. PITA and miRDB predicted potential binding sites for miR-92a-3p, miR-26a-5p, miR-23a-3p, miR-103a-3p, and miR-181a-5p in the SARS-CoV-2 3′ UTR (Figure 3B). Next, we performed qPCR analysis to determine whether these five miRNAs were present in EVs (Figure 3C). The results confirm that EVs expressed these miRNAs. To demonstrate that the five miRNAs directly bind the SARS-CoV-2 3′ UTR and suppress the RNA replication, a luciferase reporter assay was developed. The SARS-CoV-2 3′ UTR region was cloned into a luciferase reporter plasmid between the luciferase ORF and the synthetic poly(A) sequence, named pGL3 SARS-CoV-2 3′ UTR-Luc (pGL3-Luc) (Figure 3D). The recombinant plasmid was transfected into human neuroblastoma SK-N-BE(2)C cells, and luciferase activity was measured 48 h later. As shown in Figure 3E, relative luciferase activity in SK-N-BE(2)C cells transfected with the recombinant plasmids was significantly lower than that in cells transfected with the empty psi-control vector (*ps* < 0.001). Specifically, relative luciferase activity was downregulated when 3′ UTR was co-transfected along with expression vectors containing each of the five miRNAs or a vector containing all five miRNAs. The results show a significant decrease in the luciferase activity mediated by the SARS-CoV-2 3′ UTR sequence (Figure 3E). These data clearly indicate that each of the five miRNAs was able to silence the SARS-CoV-2 3′ UTR. To further investigate EVs containing functional miRNAs as capable of exerting biological effects within target cells, a luciferase assay was performed using mutant 3′ UTRs of SARS-CoV-2 (Mut). The mutated sequence was generated by a randomly designed sequence of the miRNA binding site in a reference 3′ UTR sequence. The 3′ UTR of SARS-CoV-2 (WT) or the mutated 3′ UTR of SARS-CoV-2 (Mut) were inserted into the luciferase reporter vector. For a luciferase assay, WT or Mut luciferase vectors were transfected into SK-N-BE(2)C cells and then treated with pMSC-EVs (0, 5, 10 μg) (Figure 3F). The added pMSC-EVs significantly suppressed luciferase activity of the WT-3′ UTR-containing vector in a dose-dependent manner but did not inhibit the activity of the Mut luciferase vector in SK-N-BE(2)C cells, demonstrating that EV-mediated inhibition of luciferase activity is probably the result of the specific SARS-CoV-2 3′ UTR sequence matching miRNAs. Taken together, the results suggest that all five miRNAs bind directly to the 3′ UTR of SARS-CoV-2 and suppress expression of the SARS-CoV-2 virus and either each of miR-92a-3p, miR-26a-5p, miR-23a-3p, miR-103a-3p, and miR-181a-5p or all of them are regarded as potential therapeutic agents for SARS-CoV-2.

Enrichment analysis of biological processes targeted by the five miRNAs in EVs is shown in Figure 3G. Genes related to transcription and immune regulation (including anti-inflammatory genes) are shown as blue and green circles, respectively, and genes with overlapping contributions to transcription and immune regulation are shown as red circles (Figure 3G). Interestingly, all five miRNAs had dual roles in transcription and immune regulation, further confirming that they have the potential to both degrade SARS-CoV-2 via the direct binding with the 3′ UTR and regulate the inflammatory environment created by viral infection. Based on the enrichment analysis of the five miRNA target genes in Figure 3G, the five miRNAs had the function of immune regulation. We further examined whether each of the five miRNAs within the EVs regulate pro-inflammatory cytokine release in response to SARS-CoV-2. LL24 and Beas-2B were exposed to either each of five or all of miRNAs, followed by LPS treatment. Expression of pro-inflammatory cytokines was measured. Only miR-181a-5p exhibited a significant reduction in IL-1β in Beas-2B (Figure 3H, *p* < 0.0003), and IL-6 in LL-24 cells (Figure 3H, *p* < 0.0005); the other four miRNAs did not reduce expression of pro-inflammatory factors. Interestingly, the increase in TNF-α induced by the inflammatory response was marginally significantly reduced by miR-181a-5p in Beas-2B (Figure 3H, *p* < 0.01) and in LL-24 (Figure 3I, *p* < 0.06). Furthermore, we found transfection of miR-181a-5p reduced NF-κB translocation to the nucleus markedly in Beas-2B cells (Figure 3J). Next, we analyzed the processes and pathways targeted by the five miRNAs. The mirTarBase predicted that the five miRNAs targeted 2698 genes. GO term analysis indicated that the functions of the targeted genes involved both positive and negative regulation of transcription from RNA polymerase II promoters, as well as other transcription-related functions (Appendix A). KEGG pathway analysis of the 2698 targets revealed significant enrichment of pathways related to aging (cell cycle, p53) and inflammation (PI3K, Wnt, TGF). We detected significant enrichment in pathways related to inflammatory cytokines such as TNF-α, cytokine–cytokine receptors, and chemokine receptors.

Interestingly, RNA degradation, RNA transport, protein processing in the ER, and TGF-β were also enriched (Appendix A). By preventing suppression of RNA transport, which is a survival strategy for some viruses, the five miRNAs might prevent formation of an environment favorable for viral replication. The RNA degradation pathway mainly targets genes related to the CCR4 NOT complex, which is involved in RNA homeostasis and removal of unnecessary mRNA from cytoplasmic exosomes. Given that inflammation-specific mRNAs are not cleared from CNOT3 knockout mice [29], the five miRNAs may stimulate clearance by targeting those RNAs for degradation (Appendix A). Upon infection with hepatitis C virus, ER stress markers are upregulated and proapoptotic markers are expressed [30]. The five miRNAs regulated expression of genes related to ER stress caused by virus infection (Appendix A). Influenza A virus increases expression of TGF-β and promotes cell proliferation, thereby increasing the number of cells expressing integrin subunit α5 and fibronectin, to which bacteria adhere. In addition, the innate immune system suppresses expression of the TGF-beta receptor and the SMAD pathway via IRF3 during virus infection [31,32]. By preventing suppression of this pathway, the five miRNAs normalized the local environment around virus-infected cells (Appendix A). Of the 2698 targets, we selected 83 genes associated with the GO term “inflammatory response”, and then performed KEGG pathway analysis to further identify the pathways in which they are involved; these were interleukin production and regulation, cell chemotaxis, and response to external stress (Appendix A). Moreover, the 83 genes were implicated in signal pathways related to cytokine production (e.g., the NF-κB pathway, the JNK pathway, and the Toll-like receptor pathway) (Appendix A). In addition, they target important inflammation-inducing factors such as IL-6, IL-8, and COX2. These data indicate that the five miRNAs regulate cytokine-mediated inflammatory responses and inflammatory cell activation, as well as pathways related to cytokine secretion, thereby dampening the inflammatory environment. Taken together, the above data suggest that the five miRNAs regulate the inflammatory environment and degrade viral genomic RNA through direct binding to the 3′ UTR of SARS-CoV-2.

### 3.4. The 3′ UTR Region of Several Coronaviruses Is Highly Conserved Even in Recent SARS-CoV-2 Mutations

Due to the highly conserved 3′ UTR sequence among coronaviruses, functional miRNA target sites may exist in all coronaviruses. To test this hypothesis, we selected 3′ UTR sequences from five random coronaviruses, a Sarbecovirus lineage ranging from human-infecting SARS and SARS2 virus to other bat coronaviruses, and compared them using the MUSCLE tool [28]. The sequence of SARS-CoV-2 3′ UTR of SARS-CoV-2 was aligned against the other coronaviruses. The red boxes indicate miRNA binding sites and sites predicted to bind the five miRNAs were also highly conserved (Figure 4A).

To order to study how often the SARS-CoV-2 genomic mutation also occurs in the miRNA binding site of the 3′ UTR, the site of the 3′ UTR sequence in the recently generated mutant SARS-CoV-2 was obtained from data reported in a recent paper [33]. Mutation of the 3′ UTR was found rarely in SARS-CoV-2 samples collected from China and Japan, whereas samples from Australia showed deletion of the 3′ UTR (Figure 4B). Among the mutations, five cases with 3′ UTR mutation were detected and the most common type mutation was single nucleotide mutation: 29,705 G > T, 29,854 C > T, 29,856 T > A, 29,869 deletion, and 29,749—29,759 deletion. Whereas the binding sites of hsa-miR-181a-5p, hsa-miR-92a-3p, hsa-miR-92b-3p, hsa-miR-25-3p, and hsa-miR-375-3p were mutated in the SARS-CoV-2 samples from Australia, most miRNA binding sites within the SARS-CoV-2 3′ UTR were not mutated. The hsa-miR-92a-3p is exceptional because the 3′ UTR contains two binding sites for it and one of them was reported to harbor none of the mutations within any of the variants in China, Japan, and Australia. These results suggest that most miRNAs are rarely affected by known 3′ UTR mutations (Figure 4B). Next, to investigate the SARS-CoV-2 complete genome from Korean patients, we used NCBI GenBank. Mutation of 3′ UTR was not detected in six complete genomes of SARS-CoV-2 isolated from Korean patients which were recently submitted to GenBank from February until July 2020. (Figure 4C). In summary, considering the 3′ UTR of various coronaviruses rarely possess mutations, it is highly possible that the five selected miRNAs maintain the potential to suppress all coronaviruses, including those arising via mutation. Furthermore, several miRNAs in MSC-EVs interact with other viruses. Specifically, miR-150-5p, miR-223-3p, and miR-29a-3p suppress HIV translation and prolong its latency in T cells. In addition, Let-7c-5p reduces expression of matrix protein, which is critical for influenza virus, and miR-23b-3p blocks translation of Enterovirus 71 (EV-A71). miR-181a-5p, miR-181b-5p, miR-23a-3p, miR-23b-3p, and miR-378a-3p degrade porcine reproductive and respiratory syndrome virus, whereas miR-122-5p and let-7c-5p suppress hepatitis C virus and bovine viral diarrhea virus, respectively (Figure 4D). Consequently, the presence of miRNAs in MSC-EVs that are capable of attacking various RNA-based viruses suggests that MSC-EVs can be used as a broad intervention to treat and/or to prevent virus infection.

Figure 5 shows the hypothesis, which asserts the antiviral effect of miRNA in EVs and EVs against SARS-CoV-2, and explains how miRNA directly degrades viruses and indirectly suppresses excessive immune responses. We demonstrated that key miRNAs expressed in MSC-EVs degrade SARS-CoV-2 RNAs by interacting directly with the 3′ UTR. In addition, the miRNAs in EVs exerted an anti-inflammatory effect, which prevented the cytokine storms by dampening the excessive immune response caused by the virus (Figure 5, left panel). Specifically, the direct effects of EV miRNAs against SARS-CoV-2 virus regulation are mediated by targeting regions within the SARS-CoV-2 genome, including the 3′ UTR, the 5’ UTR, and coding sequences. Particularly, direct binding to the 3′ UTR is predicted to downregulate SARS-CoV-2 RNA. In addition, EVs regenerate damaged tissue and regulate the pro-inflammatory environment via their miRNAs and protein cargoes, indicating their potential to suppress cytokine storms caused by viral infection (Figure 5, right panel). Cargoes including miRNAs in MSC-EVs attenuate induced inflammation and apoptosis caused by SARS-CoV-2, and suppress the expression of transcription/translation machinery involved in virus replication and translation, thereby indirectly suppressing the action of virus.

## 4. Discussion

Currently, viral vector vaccines are being developed by various companies, and people are actively being vaccinated worldwide. SARS-CoV-2 is an RNA virus and as such is likely to undergo frequent mutation. Therefore, vaccines or treatments must also be effective against newly arising mutant viruses. SARS-CoV-2 is an RNA virus and as such is likely to undergo frequent mutation. Therefore, vaccines or treatments must also be effective against newly arising mutant viruses. Here, we demonstrate that each individual (or all five together) miRNA in EVs suppress SARS-CoV-2 by binding directly to the 3′ UTR, resulting in translational repression (Figure 2 and Figure 3F). EVs prevent cytokine storms by normalizing inflammatory responses. Moreover, EVs or miRNAs modulate host factors to normalize immunopathogenesis. More specifically, 84 miRNAs present in EVs exerted potent indirect effects by targeting mRNAs that stimulate the immune response. Binding of target pro-inflammatory mRNAs, along with the anti-inflammatory factors contained in EVs, could normalize the immune environment, and prevent cytokine storms. In addition to the 3′ UTR region, coding sequences (CDS) within the viral genome also contain putative miRNA binding sites. Indeed, we identified the binding site of miRNAs within not only the 3′ UTR portion but also the coding region (Table 1). Several studies reported that the target inhibition is possible even when miRNAs are bound to the CDS region of the virus, but the efficiency is much lower than when binding to 3′ UTR of virus genome [34].

miRNAs can regulate the replication and translation machinery involved in virus replication and inhibit viral infection by controlling inflammation and apoptosis, both of which promote the spread of assembled virus particles [35]. Because exogenous miRNA is reported to be stable and regulate its target in vivo [36], the development of new delivery systems for miRNAs is actively being conducted in order to improve accuracy of tissue and/or cell targeting and protection from degradation by nuclease as well. To address these issues, some groups have tried modifying miRNAs. However, when such modifications are made, they create other problems, including off-target effects, reduction in miRNA activity, and formation of toxic metabolites due to miRNA degradation [37]. Consequently, an miRNA (or small RNA)-based treatment was not released until 2018, when the FDA approved a drug called Patisiran (ONPATTRO, NCT01960348). Because EVs are natural vesicles, they are safer and more effective than artificial alternatives.

Therapeutic agents based on miRNAs derived from EVs have several advantages as therapeutic agents. First, a major problem with conventional treatments for RNA-based viruses is that they have potentially fatal side effects [38]. Due to specific binding to the 3′ UTR or to the SARS-CoV-2 genome, miRNAs derived from EVs are unlikely to have significant side effects. Therefore, it is necessary to overcome the side effects of existing antiviral agents and to develop more virus-specific drugs. Moreover, because mutations in the 3′ UTR of the virus are very rare, this approach has the potential to become a universal treatment for novel RNA viruses arising via mutation of the viral RNA. Koyama et al. analyzed 10,022 cases of SARS-CoV-2 genome sequences and reported that there are more than 100 mutations in the genome sequences. Surprisingly, only two mutations were found corresponding to the 3′ UTR region, indicating that the 3′ UTR is relatively well preserved even if the SARS-CoV-2 mutation occurs. These results support our assertion that the 3′ UTR of SARS-CoV-2 virus is a target for an exogenous miRNA therapy to cope with various coronavirus variants. Our study demonstrates that the 3′ UTR sequence is highly conserved across the coronavirus family (Figure 4A), and that binding sites predicted to bind the five miRNAs are also highly conserved (Figure 4A,B). These results give us confidence that the five selected miRNAs will be effective against novel viruses arising though mutation. Third, in addition to the direct antiviral effects of miRNA, MSC-EVs can transport various cargoes that can regulate immunity and promote regeneration of damaged tissue.

MSCs have attracted attention as COVID-19 therapeutics because several lines of evidence show that they can significantly improve the pathological state of damaged lungs, including pneumonia; in addition, they could activate phagocytosis by macrophages, thereby preventing virus spread [39]. Furthermore, because MSC-EVs contain factors that can regenerate stem cells, they exhibit some of the biological properties of MSCs, and express surface receptors, signal transduction molecules, cell adhesion molecules, miRNAs, and antigens characteristic of MSCs [40,41], they will have effects against pneumonia like those of MSCs [42]. In general, it is well known that MSCs regulate the function of immune cells and accordingly are considered potential treatments for autoimmune and inflammatory diseases. According to numerous experimental and clinical studies, most MSC-based immunomodulatory functions can be attributed to the immunomodulatory properties of the EVs that they secrete. MSC-EVs are rich in biologically active molecules such as mRNA, miRNA, cytokines, and immune modulators, which regulate the function, phenotype, and viability of immune cells [43]. MSC-EVs with immune regulatory activity are also used to study degenerative, chronic, and infectious diseases. For example, when pigs infected with the influenza virus were treated with MSC-EVs, virus outflow (as detected by nasal swab) decreased, virus replication in the lungs diminished, and pro-inflammatory cytokine production was reduced significantly [44]. Because host immune responses in severe COVID-19 patients are exacerbated by pro-inflammatory factors such as IL-1, IL-6, IL-8, and impaired type I interferon activity, which can trigger a potentially fatal cytokine storm, a hallmark of severe COVID-19 according to several reports, they provide a rationale for therapeutic approaches based on EVs. In addition, a recent study reported that a multi-system inflammatory syndrome in children infected with SARS-CoV-2 results in life-threatening disease. Therefore, it is very critical to control the inflammatory syndrome before and after virus infection. Indeed, our study demonstrates that EVs from MSCs and various placental derivatives successfully inhibit secretion of pro-inflammatory factors (Figure 2 and Figure 3, Appendix A). The genes targeted by each of the top miRNAs contribute substantially to specific inflammatory responses (Figure 2E) related to cytokine–cytokine receptor interactions, TNF secretion, NF-κB activation, chemokine secretion, Toll-like receptor expression, and the Jak–STAT signaling pathway, indicating that their key function is immunomodulation (Appendix A). Specifically, miR-let-7b-5b, miR-21-5p, miR-92a-3p, miR-24-3p, and miR-181a-5p bind the most targets related to inflammatory responses.

MSC-EVs have several other advantages over MSCs. First, MSC-EVs are easier to manipulate and store [45]. Additionally, because MSC-EVs are cell-free and nano-sized, they are far less likely to trigger immunogenicity, tumorigenicity, or embolism. Because they are essentially liposomes, EVs are simple biological structures that are more stable in vivo than other particles [46]. Moreover, EVs invade host cells directly through membrane fusion, receptor-mediated phagocytosis, and other internalization mechanisms, which contribute to physiological and pathological processes such as activation of signaling pathways and immune responses [47]. Taken together, the observations reported herein suggest that MSC-EVs are an effective antiviral treatment.

More interestingly, some of the miRNAs in EVs are possible therapeutic agents for other RNA viruses (Figure 4B). Miravirsen illustrated that miRNAs that interact directly with RNA viral genomes can be used to treat hepatitis C infection; indeed, the drug has successfully completed a phase II clinical trial. Moreover, because various miRNAs present in MSC-EVs can be used to target RNA genomes with complementary sites in their 3′ UTRs, they could be used to treat infections caused by other coronaviruses, hepatitis viruses, and HIV-based RNA viruses (Figure 4C).

Although MSC-EVs are expected to be effective for antiviral therapy and immune enhancement, one major obstacle to their widespread use is that they are difficult to mass-produce. To mass-produce EVs consistently, it will be necessary to establish reproducible culture conditions and scale up the system in an economically viable manner. Moreover, mass production of EVs is limited by the challenge of producing MSCs themselves. However, in this study, we showed that the miRNA profiles of MSC-EVs and placental derivatives EVs are very similar. Consequently, it should be possible to obtain more EVs than cells by isolating placenta-derived EVs. Alternatively, it might be possible to synthesize effective miRNAs and package them into EVs to increase treatment efficacy.

To summarize, we demonstrated that EVs and five major miRNAs significantly inhibit SARS-CoV-2 replication and exert anti-inflammatory activity in vitro. Moreover, EVs regulate the pro-inflammatory environment induced by viral infection and suppress replication of SARS-CoV-2. In addition to the antiviral effect of EVs, EVs and miRNAs prevented spread of the virus when administered prior to a pro-inflammatory agent (LPS).

## Figures and Tables

**Figure 1 cells-10-02393-f001:**
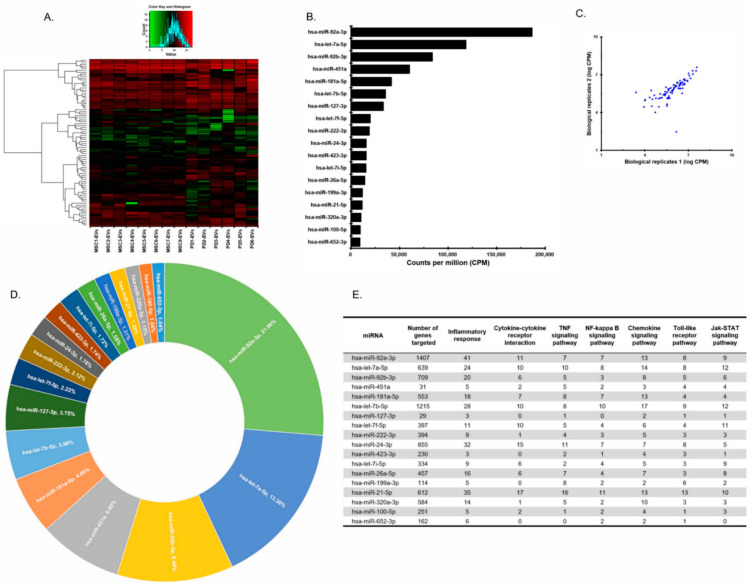
Profiles of miRNAs of pMSC-EVs and placental EVs. (**A**) The miRNA profiles of eight MSCs and six placenta-derived EVs (PD-EVs). (**B**) pMSC-EVs miRNAs were ranked (highest to lowest) in terms of counts per million (CPM). (**C**) A strong correlation between biological replicates. (**D**) The percentage of the top 18 miRNAs accounted for all EV miRNAs. (**E**) The total of 5380 genes that are predicted to be targeted by 18 miRNAs in a validated miRNA target database (the mirTarBase) and a breakdown of over-represented pathways and biological processes is shown.

**Figure 2 cells-10-02393-f002:**
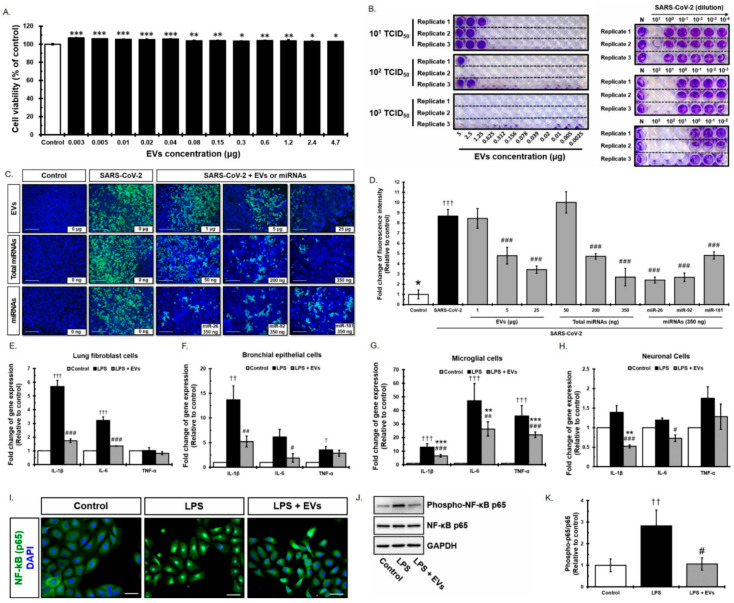
Antiviral effect of placenta-derived EVs against SARS-CoV-2. (**A**) The cytotoxicity of EVs evaluated by using the MTT assay on Vero cells. (**B**) The anti SARS-CoV-2 activity of EVs was tested using an established cell-based screening assay. Vero cells were infected with SARS-CoV-2 virus 10-fold serial dilution: 10^1^ TCID_50_/well, 10^2^ TCID_50_/well, and 10^3^ TCID_50_/well. (**C**) Antiviral effect of placenta-derived EVs, total miRNAs isolated from placenta-derived EVs, and miR-26a-5p, miR-92a-3p, miR-181a-5p against SARS-CoV-2 virus shown in immunofluorescence image. Scale bar: 200 μm. (**D**) Quantification of SARS-CoV-2 infection cells. (**E**–**H**) Protection effect of EVs on LPS stimulation in lung fibroblast cells (LL24), bronchial epithelial cells (Beas-2B), microglial cells (BV2), and neuronal cells (NSCs). (**I**) Location of NF-κB p65 after LPS treatment or LPS + EVs treatment. Scale bar: 50 μm. (**J**) Representative expression of NF-κB p65 and the expression of phospho-p65 after LPS or LPS + EVs treatment measured by Western blot analysis. (**K**) Quantification of the relative density (**J**) of the band of total NF-κB p65 and phospho-p65 (p-p65). *, **, and *** indicate *p* < 0.05, <0.01, and <0.001, respectively, compared with the EV treatment group. ^†, ††,^ and ^†††^ indicate *p* < 0.05, <0.01, and <0.001, respectively, compared with the LPS treatment group. ^#, ##,^ and ^###^ indicate *p* < 0.05, <0.01, and <0.001, respectively, compared with between LPS and EVs treated groups.

**Figure 3 cells-10-02393-f003:**
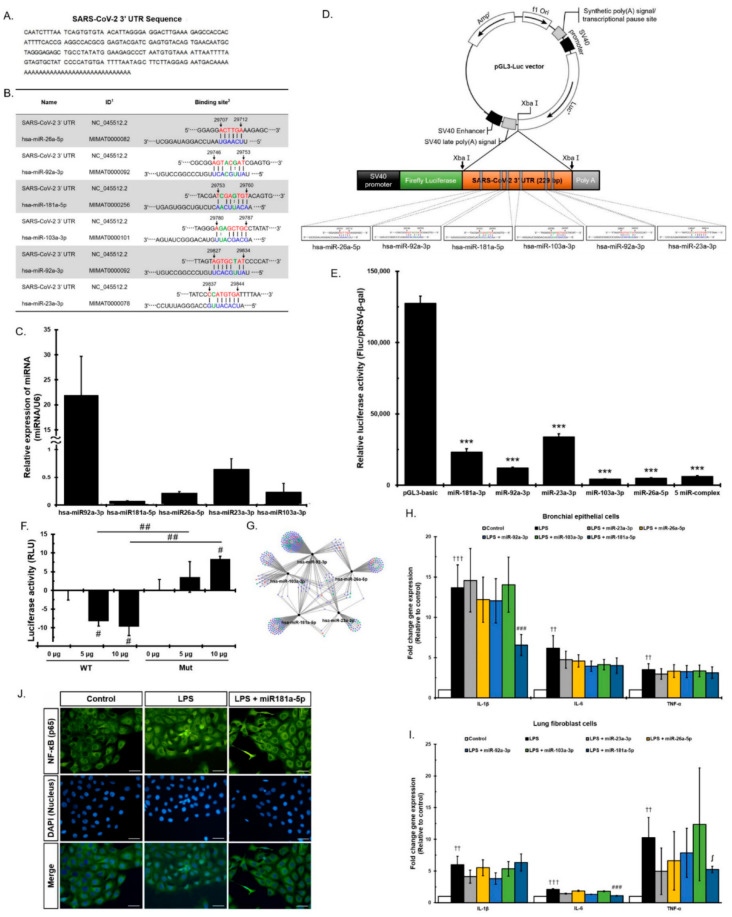
Direct viral effect of miRNAs in EVs on SARS-CoV-2. (**A**) The 3′ UTR sequence of SARS-CoV-2. The sequence was obtained from the NCBI database (accession NC_045512.2). (**B**) SARS-CoV-2 3′ UTR binding sites of five miRNAs. Each number indicates the following: 1. Accession ID of NCBI or miRBase; 2. The interaction diagram of binding site in 3′ UTR of SARS-CoV-2. A line means a perfect match, and a dot line means G:U wobble pairs. An empty space in seed regions means mismatch. (**C**) Confirmation of expression of five miRNAs measured by qPCR. (**D**) Schematic representation of the luciferase constructs used for the reporter assays. Specifically, the pGL3 SARS-CoV-2 3′ UTR construct was transfected into human neuroblastoma SK-N-BE(2)C cells. Co-transfection of these cells with miR-92a-3p, miR-26a-5p, miR-23a-3p, miR-103a-3p, miR-181a-5p, or all five miRNAs together, reduced luciferase activity. (**E**) The SARS-CoV-2 3′ UTR is a putative target of the five miRNAs identified in this study. The luciferase reporter assay revealed that miR-92a-3p, miR-26a-5p, miR-23a-3p, miR-103a-3p, and miR-181a-5p are potential regulators of SARS-CoV-2. (**F**) Either wild (WT) or mutant (Mut) forms of SARS-CoV-2 3′ UTR reporter vector were transfected to the SK-N-BE(2)C cells followed by pMSC-EVs (0, 5, 10 μg) treatment. Each group of luciferase activity is presented as percentages of difference to each mean of baseline (WT 0 μg and Mut 0 μg). (**G**) Enrichment analysis of biological processes targeted by the five miRNAs in EVs. The blue circle represents genes related to transcription, the green circle indicates immune regulation function, and the red circle shows genes with dual functions of transcription and immune regulation. (**H**) Expression of IL-1β, IL-6, and TNF-α. Bronchial epithelial cells were treated with each miRNA, followed by LPS. (**I**) Expression of IL-1β, IL-6, and TNF-α. Lung fibroblast cells were treated with each miRNA, followed by LPS. (**J**) miR-181a-5p treatment dramatically reduces NF-κB translocation to nucleus. Scale bar: 50 μm, ∫ and *** indicate *p* < 0.06, and <0.001, respectively, compared with EVs treatment group. ^††^ and ^†††^ indicate *p* < 0.01, and <0.001, respectively, compared with LPS treatment group. ^#, ##,^ and ^###^ indicate *p* < 0.05, <0.01, and <0.001, respectively, compared with between LPS and EVS treated groups.

**Figure 4 cells-10-02393-f004:**
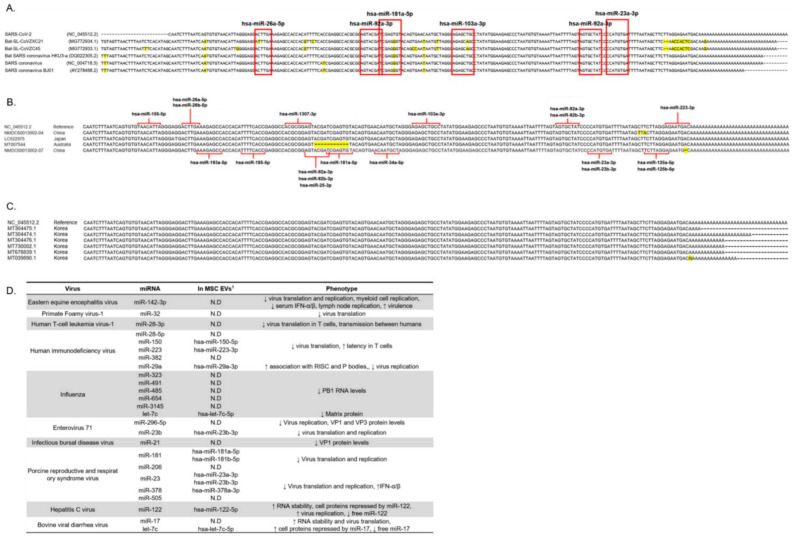
Multi-binding miRNAs will be able to target new mutant SARS-CoV-2 variants. (**A**) 3′ UTR sequence of SARS-CoV-2 of selected five coronaviruses. The red boxes indicate miRNA binding sites to the conserved regions within the 3′ UTRs of five coronaviruses. The label on the left panel shows virus names and GenBank accession. Mutated nucleotides are indicated in yellow. (**B**) Each 3′ UTR sequence was aligned to 3′ UTR sequence of NC_045512.2 using the MUSCLE tool. The label on the left panel shows GenBank accession and location where the virus was found. Red lines indicate interaction between 3′ UTR sequence and miRNA. Mutated nucleotides are indicated in yellow. (**C**) 3′ UTR sequence of 6 complete genomes obtained from SARS-CoV-2 isolated from Korean patients was aligned. Six complete genomes of SARS-CoV-2 isolated from Korean patients were submitted to GenBank from February until July 2020. Mutated nucleotides are indicated in yellow. (**D**) Modified table obtained from published reports [3]. The table shows the known interaction between miRNA and viral RNA and that there are several miRNAs existing in pMSC-EVs (1. The detected miRNAs in pMSC-EVs). “↑” means up-regulation. “↓” means down-regulation. “N.D.” means not detected.

**Figure 5 cells-10-02393-f005:**
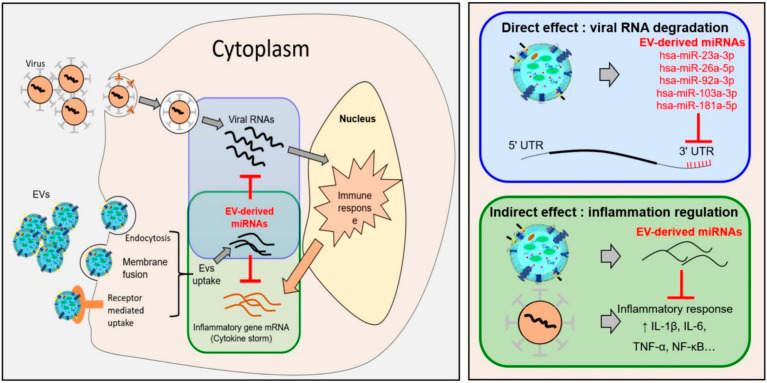
A theoretical mechanism underlying the antiviral effect of MSC-EV-derived miRNAs against SARS-CoV-2. Antiviral effect of EVs against SARS-CoV-2 viral infections causes a variety of cellular reactions, and viruses themselves are known to create an environment that allows for replication and spread. In addition, the virus causes apoptosis, and the completed virus is easily transmitted. In addition, host cells are known to secrete various cytokines and stimulate immune cells by their protective action. Consequently, it is very important to break down viral replication before stimulation of inflammatory cytokines. Our hypothesis of antiviral effects of miRNAs in EVs and EVs against SARS-CoV-2: SARS-CoV-2 viruses invade cells through ACE2 receptors and EVs enter cells through various pathways, including membrane fusion, receptor-mediated uptake, and active endocytosis. The cell secretes cytokines, induces an immune fraction, and stimulates immune cells to fight the viral infection. Too much cytokine leads to a cytokine storm (**left** panel). The direct effects of EV miRNAs on SARS-CoV-2 virus regulation are mediated by targeting regions within the SARS-CoV-2 genome, including the 3′ UTR, the 5’ UTR, and coding sequences. In particular, direct binding to the 3′ UTR is predicted to downregulate SARS-CoV-2 RNA. In addition, EVs regenerate damaged tissue and regulate the proinflammatory environment via their miRNAs and protein cargoes, indicating their potential to suppress cytokine storms caused by viral infection (**right** panel). “↑” means up-regulation.

**Table 1 cells-10-02393-t001:** miRNAs predicted to bind the SARS-CoV-2 RNA genome.

PITA 3’UTR Binding Prediction ^1^							mirDB Unconventional Target Sites Prediction ^2^		
microRNA	Expression (%) ^3^	Seed Location ^4^	Seed Match Length ^5^	Mismatch ^6^	G:U Wobble ^7^	microRNA-Target Hybridization Energy ^8^	Score ^9^	Seed Location ^10^	Rank ^11^
hsa-miR-92a-3p	22.23476	29,746	8	1	1	−13.8	−	−	1
hsa-miR-92b-3p	9.978815	29,746	8	1	1	−18.8	−	−	3
hsa-miR-181a-5p	4.92776	29,753	8	1	1	−18.7	70	7410, 7529, 8221, 9016, 11,400, 12,216, 18,516, 20,783, 27,948	5
hsa-miR-26a-5p	1.672265	29,707	6	0	0	−14.9	68	454, 9596, 20,513, 27,848, 29,707	12
hsa-miR-34a-5p	0.925787	29,768	8	1	1	−13.25	−	−	20
hsa-miR-23a-3p	0.790515	29,837	8	1	0	−8.1	79	6458, 7908, 15,302, 21,244	23
hsa-miR-125b-5p	0.389675	29,856	8	1	1	−11.9	−	−	36
hsa-miR-125a-5p	0.324866	29,856	8	1	1	−10.4	−	−	37
hsa-miR-103a-3p	0.269252	29,780	8	1	1	−12.6	85	8827, 13,089, 14,561, 14,780, 22,345, 24,235, 25,319, 26,371, 27,101, 28,734, 28,920, 29,461	39
hsa-miR-223-3p	0.233216	29,863	8	1	1	−5.6	−	−	45
hsa-miR-25-3p	0.153562	29,746	8	1	1	−15.5	−	−	51
hsa-miR-26b-5p	0.149508	29,707	6	0	0	−11.3	68	454, 9596, 20,513, 27,848, 29,707	52
hsa-miR-193a-5p	0.130145	29,712	8	1	0	−12.1	−	−	57
hsa-miR-1307-3p	0.129069	29,740	8	1	0	−30.9	−	−	58
hsa-miR-155-5p	0.098556	29,693	8	1	0	−8.3	51	864, 5209, 17,197, 25,074	63
hsa-miR-185-5p	0.050131	29,726	8	1	1	−11.31	−	−	69
hsa-miR-23b-3p	0.02939	29,837	8	1	0	−11.3	79	6458, 7908, 15,302, 21,244	74

Each number indicates following: 1. Predictions of 3′ UTR binding were conducted with the PITA tool. 2. To predict miRNA binding to the 3′ UTR and region, analysis of the whole genome was also performed using miRDB. 3. Expression (%) indicates the level of each miRNA relative to the total miRNA levels, calculated miRNA CPM/total CPM × 100. 4. Seed location where each miRNA is expected to bind the SARS-CoV-2 genome. 5. Seed match length. 6. Seed mismatch length. 7. Wobble number caused by G and U base pairing. 8. Thermodynamic energy required for binding. Lower energy indicates stronger binding prediction. The higher the score, the greater the likelihood of strong binding. 9.The prediction score from the miRDB method. According to miRDB, a predicted target with score >80 is most likely to be tightly bound. 10. Start site of the miRNA seed location matching to the RNA genome of SARS-CoV-2. 11. Rank of expression level within EVs.

## Data Availability

Not applicable.

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
