# Peer review of "Potential Therapeutic Effect of Micrornas in Extracellular Vesicles from Mesenchymal Stem Cells against SARS-CoV-2"

_cells, 2021, doi:10.3390/cells10092393_

Round 1
Reviewer 1 Report
Broad comments
This manuscript may provide novel insights in an antiviral role of adult stem cells such as MSC and specifically on placenta derived cells. The viral target, the new coronavirus, as subject of the study should be important but carefully considered by the scientific editors and be restricted to non-directly translational evidence but as basic research tools after publication. The experiments are clean and no big issues were found, however the organization of the manuscript and many points must be clarified or improved for this reviewer and in order to publish on this high quality journal which is Cells (by MDPI)
Major points
1)The abstract seems more appropriate to a review article rather than a research article.
Abstract can revised. Some current parts can be shortened and instead experimental results must be briefly added. Take into consideration the limit of words that journal instructions for
Authors reports.
2)About the introduction section. Vaccine takes time but it is universally considered the best weapon against this type of viral infection, thus this sentence can be adjusted and the limitations should not highlight a negative aspect in comparison to other treatments . Solve the partial contradiction, please
3)The authors wrote” RNAs as their genome and are frequently susceptible to mutation”. So, it would be nice to explain how and why ev-msc will be useful in case of mutations and viral variants. Does the binding to the 3' UTR of SARS-CoV-2 one of the reason? Please briefly clarify in the introduction
4)A reference for other antimicrobial and immunomodulatory effects of the extra cellular vesicles (including microvesicles and exosomes) derived from MSCs should be added to the introduction. A recent article summarize these evidences and can be found at doi.org/10.3390/antibiotics10070750. Indeed, MSCs have shown both direct and indirect antibacterial effects ( as shown by Marrazzo et al. 2019) . The addition of the above citation can be linked also to what the authors discuss from line 706 to 714.
5)How miR-26a-5p, miR-92a-3p, miR-181a-5p were identified? They appeared suddenly in the text, but I guess they were identified as described in the Figure 2. Please add this point while mentioning first time this specific miRNAs.
6)I think the manuscript will benefit the exchange between section and Figure 1 and 2 respectively. Thus section 3.1 that is very big can be split. The. Figure 2 (Profiles of miRNAs of pMSC-EVs and placenta EVs. ) should be the first (i.e. new Fig 1) of the manuscript and accompanied by respective results of current section 3.1.
7)the results obtained by the EVs basically depended on the miRNA comprised in them. However, how much this pool of effective miRNA is present in non-stem cells and other ACE-positive EVs? Reference of preliminary data or supplementary info should be added to verify the specific and useful effect of EV-MSCs against infection.
8)first line of the discussion should be changed and updated. Today, many vaccines for SARS-CoV-2 exist.
Minor points
9)Since cell lines and not primary cells were used for in vitro exploration of EV-MSC effects, a note regarding dind the limitations of using cell lines as well as the advantage of future experiments on organoids deriving from stem cells can ve useful
10)About Figure 1 legend and labels, are “total miRNAs” the combination of miR-26a-5p, miR-92a-3p, miR-181a-5p ? Please be clearer
11)Line 555, “Enterovirus 71” can also named in bracket aka A71. I suggest because this will help the reader to understand the number 71 is not a missing reference of typo.
Author Response
Ref.: Manuscript ID: cells-1354393
Dear Chief of Editor of Cells
We really appreciate the Editors in Cells and all Reviewers taking time to review and providing many brilliant insights to improve the quality of our manuscript entitled, “Potential therapeutic effect of microRNAs in extracellular vesicles from mesenchymal stem cells against SARS-CoV-2.” In response, we have carefully revised the manuscript, addressing all the raised concerns by reviewers and changes are highlighted in blue in the revised manuscript.
We respond the questions and comments point by point with pleasure and please find our point-by-point answers to the reviewers’ comments below. Changes are highlighted in blue in the revised manuscript.
Thank you again for considering this manuscript.
To reviewers,
Thank you for your time and efforts to review our manuscript in detail and for pointing out many important issues that can strengthen our manuscript. We totally agree with your opinions and revise the manuscript.
Reviewer #1:
Major points
1)The abstract seems more appropriate to a review article rather than a research article.
Abstract can revised. Some current parts can be shortened and instead experimental results must be briefly added. Take into consideration the limit of words that journal instructions for
We appreciate your advice and admit oversight and we have improved the abstract as suggested:
Extracellular vesicles (EVs) are cell-released, nanometer scaled, membrane bound materials and contain diverse contents including proteins, small peptides, and nucleic acids. Once released, EVs can alter microenvironment and regulate a myriad of cellular physiology including cell-cell communication, proliferation, differentiation, and immune responses against viral infection. Among the cargos in the vesicles, small non-coding micro-RNAs (miRNAs) have received attention in that they can regulate expression of a variety of human genes as well as external viral genes via binding to the complementary mRNAs. In this study, we tested the potential of EVs as therapeutic agents for severe acute respiratory syndrome coronavirus 2 (SARS-CoV-2) infection. First, we found that the mesenchymal stem cell derived-EVs (MSC-EVs) enabled to rescue the cytopathic effect of SARS-CoV-2 virus and to suppress proinflammatory responses in the infected cells by inhibiting the viral replication. We found that these anti-viral responses were mediated by 17 miRNAs matching the rarely mutated, conserved 3’-untranslated regions (UTR) of the viral genome. The top five miRNAs highly expressed in the MSC-EVs: miR-92a-3p, miR-26a-5p, miR-23a-3p, miR-103a-3p, and miR-181a-5p were tested. They were bound to the complemented sequence which led recovery of the cytopathic effects. These findings suggest that the MSC-EVs are a potential candidate for multiple variants of anti-SARS-CoV-2.
Authors reports.
2)About the introduction section. Vaccine takes time but it is universally considered the best weapon against this type of viral infection, thus this sentence can be adjusted and the limitations should not highlight a negative aspect in comparison to other treatments. Solve the partial contradiction, please
We are grateful for the suggestion for enhancing the introduction. We have updated the introduction as following.
Many biotech companies have focused on the development of vaccines against COVID-19. Because multiple trials of safety and efficacy tests remain, the more efficacy of vaccines should be awaited to prevent or reduce the viral infection. Under these circumstances, the development of therapeutic agents for COVID-19 is in progress, suggesting that two research tracks should be pursued: vaccine and drug development to overcome infectious diseases (line 44~49).
3)The authors wrote” RNAs as their genome and are frequently susceptible to mutation”. So, it would be nice to explain how and why ev-msc will be useful in case of mutations and viral variants. Does the binding to the 3' UTR of SARS-CoV-2 one of the reason? Please briefly clarify in the introduction
We appreciate your comments and apologize for not clearly explaining why the 3' UTR of the viral genome is considered a good target for miRNAs in EVs for antiviral effects.
It is well known that 3'UTR of RNA virus is conserved (#, Ref.4, 5), and we also performed 3' UTR analysis of known coronaviruses and found that the 3' UTR region of coronaviruses was highly conserved (Fig.4) indicating that this region would not be mutated as much as the coding regions.
Based on these results, we further investigate miRNAs in EVs which are supposed to bind to the 3’ UTR of SARS-CoV-2. 17 miRNAs in the EV that bind to the 3' UTR region of SARS-CoV-2 virus were found. For further testing the anti-viral effects against the SARS-CoV-2, five highly expressed miRNAs were selected for the further experiments.
As suggested, we have been added the following to the introduction.
“The RNA genome has a high rate of mutation, making RNA viruses very heterogeneous [1]. As HIV and influenza, SARS-CoV-2 viruses contain RNAs as their genome and are frequently susceptible to mutation. New SARS-CoV-2 variants have been identified in a stream and cause doubt in development of vaccines and therapies.
For example, antibody-based therapeutics developed against one variant must be redesigned after the virus mutates. Consequently, influenza vaccines need to be updated annually. Also, viral variants develop resistance to antiviral drugs, and mild virus variants become virulent spontaneously. Consequently, therapeutic and preventive agents that can cope with viral mutations are needed urgently”. The heterogeneity features of SARS-Cov-2 asks us to find new approaches such as recognizing common regions of multiple variants and deactivating viral functions (line 50~60).
“Trobaugh D.W. et. reported that Eastern Equine Encephalitis Virus (EEEV) was inhibited by the interaction of miR-142-3p with the complementary miR-142-3p binding site in the EEEV 3' UTR region. However, removal of its 3' UTR led to reduction of viral replication, indicating that the 3‘ UTR in viral genome has its own function in viral infection [following reference #, Ref. 1, 2, 3]. Secondly, it is known that the 3' UTR region of the virus genome is relatively conserved and forms a conserved special secondary structure which is known to play an important role in virus replication as well as stabilizing RNA (following reference #, Ref.4, 5), further confirming the importance of targeting 3’ UTR of virus. Given that there is a plausible mechanism that the complementary miRNAs can potentially target the virus and, above all, exert antiviral effects by mainly acting on the miRNA binding site within the conserved 3' UTR, the targeting of 3' UTR of the viral genome can be a key strategy for antiviral effects.” (line 72~83).
1.Heiss, Brian L., et al. "MicroRNA targeting of neurotropic flavivirus: effective control of virus escape and reversion to neurovirulent phenotype." Journal of virology 86.10 (2012): 5647-5659.
2.Trobaugh, Derek W., and William B. Klimstra. "MicroRNA regulation of RNA virus replication and pathogenesis." Trends in molecular medicine 23.1 (2017): 80-93.
3.Trobaugh, Derek W., et al. "RNA viruses can hijack vertebrate microRNAs to suppress innate immunity." Nature 506.7487 (2014): 245-248.
4.Dreher, Theo W. "Functions of the 3′-untranslated regions of positive strand RNA viral genomes." Annual review of phytopathology 37.1 (1999): 151-174.
5.Yang, Dong, and Julian L. Leibowitz. "The structure and functions of coronavirus genomic 3′ and 5′ ends." Virus research 206 (2015): 120-133.
4) A reference for other antimicrobial and immunomodulatory effects of the extra cellular vesicles (including microvesicles and exosomes) derived from MSCs should be added to the introduction. A recent article summarize these evidences and can be found at doi.org/10.3390/antibiotics10070750. Indeed, MSCs have shown both direct and indirect antibacterial effects (as shown by Marrazzo et al. 2019). The addition of the above citation can be linked also to what the authors discuss from line 706 to 714.
Thank you for your comments and we have added the reference in the manuscript (line 95~96).
5) How miR-26a-5p, miR-92a-3p, miR-181a-5p were identified? They appeared suddenly in the text, but I guess they were identified as described in the Figure 2. Please add this point while mentioning first time this specific miRNAs.
We sincerely appreciate the comments of the reviewers. Thank you for carefully checking the manuscript. To account for the miRNAs mentioned first in the results, we have added explanation of the miRNAs as follows:
To understand mechanism of antiviral effects mediated by either total EVs or miRNAs in EVs, miRNAs in EVs were analysed and identified (Figure 3 and Table 1). Among the most abundant miRNAs in EVs, miR-92a-3p, miR-181a-5p, and miR-26a-5p can target the SARS-CoV2 with high efficiency and have potential to regulate inflammatory responses (line 379~381).
6) I think the manuscript will benefit the exchange between section and Figure 1 and 2 respectively. Thus section 3.1 that is very big can be split. The. Figure 2 (Profiles of miRNAs of pMSC-EVs and placenta EVs. ) should be the first (i.e. new Fig 1) of the manuscript and accompanied by respective results of current section 3.1.
Thank you for your valuable suggestion. We have rearranged figures and respective results as the reviewer suggested.
7) the results obtained by the EVs basically depended on the miRNA comprised in them. However, how much this pool of effective miRNA is present in non-stem cells and other ACE-positive EVs? Reference of preliminary data or supplementary info should be added to verify the specific and useful effect of EV-MSCs against infection.
We really appreciate your suggestions and we have been also interested in the issues because the contents in EV are quite different depending on diverse conditions such as the source, physiological condition, stresses, and age, and the variations of contents are important in choice of cells for treatment.
The reference regarding with EV-MSCs against infection has added in the manuscript (line 95~96)
8) first line of the discussion should be changed and updated. Today, many vaccines for SARS-CoV-2 exist.
Thanks for your comments. We have updated many vaccines which have developed such as:
Currently, viral vector vaccines developed by various companies and people are actively being vaccinated worldwide. SARS-CoV-2 is an RNA virus and as such is likely to undergo frequent mutation. Therefore, vaccines or treatments must also be effective against newly arising mutant viruses (line 645~648).
Minor points
9)Since cell lines and not primary cells were used for in vitro exploration of EV-MSC effects, a note regarding dind the limitations of using cell lines as well as the advantage of future experiments on organoids deriving from stem cells can ve useful
We totally agree with your comments. Whether a cell line-based experiment is an appropriate experimental model in in vitro experiments has been a constant debate.
Because we know that results from experiments of cell lines were not perfect, we have tried to conduct experiments mainly on cell lines that are already well known in the virus field. Vero cell is a cell line mainly used in experiments related to SARS-CoV-2 infection [following reference #, Ref. 1, 2, 3], A549, Beas-2B, and BV2 cells are also commonly used cell lines that have been widely used to study inflammatory responses to target organs.
Currently, in our laboratory, we are conducting research on organoids to construct more sophisticated experimental models and hope to report the results soon.
- Mantlo, Emily, et al. "Antiviral activities of type I interferons to SARS-CoV-2 infection." Antiviral research 179 (2020): 104811.
- Ogando, Natacha S., et al. "SARS-coronavirus-2 replication in Vero E6 cells: replication kinetics, rapid adaptation and cytopathology." The Journal of general virology 101.9 (2020): 925.
- Runfeng, Li, et al. "Lianhuaqingwen exerts anti-viral and anti-inflammatory activity against novel coronavirus (SARS-CoV-2)." Pharmacological research 156 (2020): 104761.
10) About Figure 1 legend and labels, are “total miRNAs” the combination of miR-26a-5p, miR-92a-3p, miR-181a-5p? Please be clearer
Thanks for your comment. We have clarified the legend and labels of Figure 1 (line 425).
11) Line 555, “Enterovirus 71” can also named in bracket aka A71. I suggest because this will help the reader to understand the number 71 is not a missing reference of typo.
Thanks again for your careful comment. We have changed in the manuscript (line 593).
Reviewer 2 Report
- The first sentence of the discussion reads as "As yet, there is no vaccine or treatment for SARS-CoV-2". Moreover, there are no articles cited published during 2021. These altogether suggest that the article has been moving from one journal to another without proper update, particularly on the novelty and implications of the findings for the current research and policy scenarios. Please, review accordingly (remarkably, the discussion and conclusions, but also the other sections of the manuscript).
- I don't feel the abstract approppriately summarizes the contents of the article, and particularly major methodological features are missing.
- There are some formal aspects to be considered:
- Line 40: reference 1 is used too late within the text, after several major statements that need evidence back up have already been layed out. Please, either cite earlier or include further (and updated references)
- Line 61: Please define EVs before first use
- Line 73: define ncRNAs
- Line 96: There is a sudden leap from reference 11 to reference 33
- Line 107: define NSCs (defined further below in line 325)
- Line 108: "gentamicin" instead of "gentamycin"
- Line 115: define qEV
- Line 117: define PBS
- Line 198: "COVID virus" incorrect (SARS-CoV-2)
- Line 300: cytopathic effect (CPE) acronym already been used and defined in line 192
- In Results, I recommend avoiding connotated and interpretative expressions such as the following in Line 317: "These results strongly suggest that EV or miRNA treatment has antiviral effects against SARS-CoV-2 infection".
- Please, consider adjusting the following subheadline in line 524: "3.2. 3'UTRs of corona virus are highly conserved"; perhaps too broad?
- Figure 4 footnote D: please revise the reference to the Trobaugh et al article.
Author Response
Ref.: Manuscript ID: cells-1354393
Dear Chief of Editor of Cells
We really appreciate the Editors in Cells and all Reviewers taking time to review and providing many brilliant insights to improve the quality of our manuscript entitled, “Potential therapeutic effect of microRNAs in extracellular vesicles from mesenchymal stem cells against SARS-CoV-2.” In response, we have carefully revised the manuscript, addressing all the raised concerns by reviewers and changes are highlighted in blue in the revised manuscript.
We respond the questions and comments point by point with pleasure and please find our point-by-point answers to the reviewers’ comments below. Changes are highlighted in blue in the revised manuscript.
Thank you again for considering this manuscript.
To reviewers,
Thank you for your time and efforts to review our manuscript in detail and for pointing out many important issues that can strengthen our manuscript. We totally agree with your opinions and revise the manuscript.
Reviewer #2:
- The first sentence of the discussion reads as "As yet, there is no vaccine or treatment for SARS-CoV-2". Moreover, there are no articles cited published during 2021. These altogether suggest that the article has been moving from one journal to another without proper update, particularly on the novelty and implications of the findings for the current research and policy scenarios. Please, review accordingly (remarkably, the discussion and conclusions, but also the other sections of the manuscript).
Thanks for your comments. We have updated many vaccines which have developed such as:
Currently, viral vector vaccines developed by various companies and people are actively being vaccinated worldwide. SARS-CoV-2 is an RNA virus and as such is likely to undergo frequent mutation. Therefore, vaccines or treatments must also be effective against newly arising mutant viruses (line 645~648).
- I don't feel the abstract approppriately summarizes the contents of the article, and particularly major methodological features are missing.
We appreciate your advice and admit oversight and we have improved the abstract as suggested:
Extracellular vesicles (EVs) are cell-released, nanometer scaled, membrane bound materials and contain diverse contents including proteins, small peptides, and nucleic acids. Once released, EVs can alter microenvironment and regulate a myriad of cellular physiology including cell-cell communication, proliferation, differentiation, and immune responses against viral infection. Among the cargos in the vesicles, small non-coding micro-RNAs (miRNAs) have received attention in that they can regulate expression of a variety of human genes as well as external viral genes via binding to the complementary mRNAs. In this study, we tested the potential of EVs as therapeutic agents for severe acute respiratory syndrome coronavirus 2 (SARS-CoV-2) infection. First, we found that the mesenchymal stem cell derived-EVs (MSC-EVs) enabled to rescue the cytopathic effect of SARS-CoV-2 virus and to suppress proinflammatory responses in the infected cells by inhibiting the viral replication. We found that these anti-viral responses were mediated by 17 miRNAs matching the rarely mutated, conserved 3’-untranslated regions (UTR) of the viral genome. The top five miRNAs highly expressed in the MSC-EVs: miR-92a-3p, miR-26a-5p, miR-23a-3p, miR-103a-3p, and miR-181a-5p were tested. They were bound to the complemented sequence which led recovery of the cytopathic effects. These findings suggest that the MSC-EVs are a potential candidate for multiple variants of anti-SARS-CoV-2.
- There are some formal aspects to be considered:
Thank you for your comments on format, mistake, and ambiguous statement. We have addressed all your valuable points.
- Line 40: reference 1 is used too late within the text, after several major statements that need evidence back up have already been layed out. Please, either cite earlier or include further (and updated references)
Changed in line 40.
- Line 61: Please define EVs before first use.
Full name of EVs were added in line 83.
- Line 73: define ncRNAs
Full name of ncRNAs were added in Line 97.
- Line 96: There is a sudden leap from reference 11 to reference 33
The reference number has been revised sequentially.
- Line 107: define NSCs (defined further below in line 325)
Full name of NSCs were added in Line 133.
- Line 108: "gentamicin" instead of "gentamycin"
Corrected a misspelling of gentamicin in line 134.
- Line 115: define qEV
Defined qEV in line 141~142.
- Line 117: define PBS.
Defined PBS in line 143~144.
- Line 198: "COVID virus" incorrect (SARS-CoV-2)
Corrected the COVID virus to SARS-CoV-2 in line 226.
- Line 300: cytopathic effect (CPE) acronym already been used and defined in line 192.
Corrected as suggested in line 362.
- In Results, I recommend avoiding connotated and interpretative expressions such as the following in Line 317: "These results strongly suggest that EV or miRNA treatment has antiviral effects against SARS-CoV-2 infection".
To avoid interpretative expressions, the sentence has been removed.
- Please, consider adjusting the following subheadline in line 524: "3.2. 3' UTRs of corona virus are highly conserved"; perhaps too broad?
The 3' UTR region of several coronaviruses is highly conserved even in recent SARS-CoV-2
Mutations in line 561~562.
- Figure 4 footnote D: please revise the reference to the Trobaugh et al article.
Revised to the reference as suggested in line 609.
Round 2
Reviewer 1 Report
The authors provided a clear, complete and kind rebuttal, thanks. The paper is improved.
The authors should and will take own responsability about the specificity of targeting 3' UTR for Coronaviruses and no other viruses.
Reviewer 2 Report
The authors have satisfactorily answered my questions and remarks. Minor English copyediting will still be required.